



**Vegetative impacts upon bedload transport capacity and**
**channel stability for differing alluvial planforms in the**
**Yellow River Source Zone**
**Z. W. Li[1], G. A. Yu[2], G. Brierley[3], and Z.Y. Wang[4]**
[1]{School of Hydraulic Engineering, Changsha University of Science & Technology,
Changsha, China}
[2]{Institute of Geographic Sciences and Natural Resources Research, Chinese Academy of
Sciences, Beijing, China}
[3]{School of Environment, University of Auckland, Auckland, New Zealand}
[4]{State Key Laboratory of Hydroscience and Engineering, Tsinghua University, Beijing,
China}
Correspondence to: G. A. Yu (yuga@igsnrr.ac.cn)
**Abstract**
The influence of vegetation upon bedload transport and channel morphodynamics is
examined along a channel stability gradient ranging from meandering through anabranching
through anabranching-braided to fully braided planform conditions along trunk and tributary
reaches of the Upper Yellow River in western China. Although the regional geology and
climate are relatively consistent across the study area, there is a distinct gradient in the
presence and abundance of riparian vegetation for these reaches atop the Qinghai-Tibet
Plateau (elevations in the study area range from 2800-3400 m a.s.l.). The hydraulic and
geomorphic role of riparian vegetation varies as follows: trees exert the strongest influence in
the anabranching reach, the meandering reach flows through meadow vegetation, the
anabranching-braided reach has a grass, herb, and sparse shrub cover, and the braided reach
has no riparian vegetation. A non-linear relation between vegetative cover on the valley floor
and bedload transport capacity is evident, wherein bedload transport capacity is highest for
the anabranching reach, followed by the anabranching-braided, braided and meandering
reaches respectively. The relationship between the bedload transport capacity of a reach and



sediment supply from upstream exerts a significant influence upon channel stability. Bedload
transport capacity during the flood season (June-September) in the braided reach is much less
than the rate of sediment supply, inducing bed aggradation and dynamic channel adjustments.
Rates of channel adjustment are less pronounced for the anabranching-braided and
anabranching reaches, while the meandering reach is relatively stable (i.e. this is a passive
meandering reach).

## 1   Introduction

Transitions in river character and behaviour are a key focal point of enquiry in fields such as
geomorphology, hydrology, and sedimentology. Such concerns have significant management
applications, especially relating to issues such as management of flood risk and sedimentation
hazards. These issues are likely to become even more pronounced in the future, as rivers
adjust in response to climate and land use changes, and management actions. Putting aside
concerns for terminological issues associated with differentiation of river types and their
morphological attributes (see Lewin and Ashworth, 2014, Carling et al., 2014, Tadaki et al.,
2014), it is clear that a concerted effort is required to generate process-based understandings
of morphodynamic adjustments to address concerns for prospective future river changes
(Beechie et al., 2010). Here we evaluate the influence of riparian vegetation upon these
process interactions, focussing upon relatively understudied reaches of the Upper Yellow
River atop the Qinghai-Tibet Plateau in western China.
Channel bars are products of instream deposition of bedload materials, whether at the channel
margin (bank-attached forms) or mid-channel features (bars) (Brierley and Fryirs, 2005).
Typically, bars mutually adjust with channel geometry, such that they scale to the size of the
channel in which they form (Task Force on Bed Forms in Alluvial Channels, 1966; Nicholas
et al., 2013). If these features become vegetated and stabilized, they are referred to as islands.
Unit bars (migrating lobate bed forms with heights and lengths that scale with channel depth
and width) are differentiated from larger, more complex compound bars (e.g. Bridge, 1993;
Brierley, 1989, 1991; Smith, 1974). Compound bars are products of multiple phases of
accretion and reworking, with stacked sequences of unit bar, dune, and smaller bed form
deposits that are often trimmed at their margins by bank erosion processes or dissected by
chute channels (Ashworth et al., 2011; Best et al., 2003; Bridge, 2003; Brierley and Fryirs,
2005; McGowen and Garner, 1970; Reesink et al., 2014; Sambrook Smith et al., 2009).



Various studies have characterized the main morphological elements of large bars and islands,
while other studies have developed conceptual models of bar evolution (e.g. Ashworth et al.,
2000; Gurnell et al., 2001, Latrubesse and Franzinelli, 2002; Mertes et al., 1996). Moreover,
Brierley and Hickin (1991) and Brierley (1996) highlight how analyses of sediment sequences
made up of facies and element-scale assemblages of bar deposits cannot be used to
differentiate among channel planform types.
There is notable variability in the presence, form and hydraulic/sedimentologic
(morphodynamic) role of bars along the continuum of channel planform (Bridge, 1993;
Brierley, 1996). By definition, as suspended-load rivers have limited bedload-calibre
materials, they have very few, if any, bars. The prominence of fine-grained (silt-clay) deposits
under low energy conditions (often very low channel gradient) promotes passive channel
behaviour, typically with a low sinuosity, passive meandering or anabranching (anastomosing)
planform (Eaton et al. 2010; Fryirs and Brierley, 2012; Makaske, 2001; Wang et al., 2005).
Patterns of bar formation in mixed- and bedload-dominated rivers reflect the flow-sediment
balance along any given reach, with a spectrum of planform types ranging from active
meandering and wandering variants through to fully braided rivers (see Ashworth, 1996;
Ashworth and Lewin, 2012; Burge, 2006; Church and Rice, 2009). Braiding results from the
inability of flow to transport all sediments that are made available to the channel, such that
mid-channel sedimentation occurs (i.e. competence and/or capacity limits are exceeded,
wherein sediment is either too coarse to be transported, or there is too much sediment for the
flow to transport, respectively). Recurrent reworking of bedload materials via thalweg shift
during flood events alters the number, shape, and location of bars. Bar dissection and avulsion
create multi-thread channel systems with a disorderly river planform, extremely unstable bars,
and inconstant flow paths (Ashmore, 1991; Ashworth et al., 2000; Jerolmack and Mohrig,
2007). However, if channel boundary conditions induce sufficient bank strength, and flows
are able to transport available bedload sediments, the river adopts a configuration with better-
defined, less mobile channels with a much lower width-depth ratio, whether within a single-
channel configuration (typically passive meandering) or a multi-channel anabranching
configuration (Eaton et al., 2010; Song and Bai, 2015). Controversy abounds in our
theoretical understanding of process controls upon anabranching river behaviour (see Carling
et al., 2014; Nicholas et al., 2013). While Huang and Nanson (2007) and Jansen and Nanson
(2004, 2010) attribute an anabranching configuration to the least action principle, wherein
channels adjust their form to transport available sediment in the most hydraulically efficient



manner, Eaton and co-workers postulate quite opposite situations in which anabranching
channels adjust to minimize their capacity to transport materials (Eaton and Church, 2004,
2007; Eaton et al., 2010). This builds upon long-standing awareness of the differing
environmental conditions under which anabranching and anastomosing planform types are
observed, ranging from tropical to arid zone settings (see Nanson and Knighton, 1996;
Nanson, 2013). This perhaps suggests that different factors can result in these channel
configurations (the principle of geomorphic convergence, or equifinality). It is not our
concern here to address this issue directly. Rather, our focus lies with analysis of relationships
between bedload transport capacity and channel morphodynamics along a continuum of
channel planform types along the Upper Yellow River. This continuum is coincident with a
gradation in riparian vegetation cover (Yu et al., 2014).
In some instances, vegetation may support the long-term stable development of sandbars
within a stable multi-channel system – a variant of an anabranching river (Latrubesse, 2008;
Nanson and Knighton, 1996; Murray and Paola, 2003; Tal and Paola, 2010). Bar stability is
the key distinguishing attribute of braided and anabranching rivers. Vegetation increases flow
resistance and stabilizes the channel bed and bank in the latter instance, thereby altering
channel geometry, bedload transport rates, and the resulting rates and patterns of bed
deposition or erosion. Once a particular morphology has been formed, the configuration of
channels and associated distribution of bars and roughness elements fashions process
responses to subsequent flood events (Hooke, 1986, 2015; Hooke and Yorke, 2011; Luchi et
al., 2010). If critical threshold conditions are exceeded, alterations to the balance and patterns
of erosion and deposition processes may bring about transitions to different planform types.
Mutual adjustments between patterns of vegetation types (size, spacing, and density) and
flow-sediment dynamics (patterns and rates of erosion and deposition) vary at different
positions on the valley floor. Vegetation encroachment by pioneer species and successional
processes induce abiotic and biotic transitions in geomorphic processes from the unvegetated
channel bed and bar surfaces to grassland, shrubs, and treed areas at the margins of
bars/islands and on floodplains (Corenblit et al., 2007, 2011; Gurnell, 2014; Hickin, 1984;
Hupp and Osterkamp, 1996; Millar, 2000; Tooth and Nanson, 2000). Vegetation attributes
influence the pattern of roughness elements and the associated distribution of flow energy,
thereby affecting the distribution of erosional and depositional processes, and resulting
morphological attributes (including the grain size distribution of bed/bar materials). Hence,





vegetative controls influence the stability and behaviour of alluvial bed and bars, and the
influence of vegetation upon flow-sediment interactions, vary for differing planform types
(Gran and Paola, 2001; Gradzinski et al., 2003; Jang and Shimizu, 2007; McBride et al.,

4   2007).

Although the prominence of seasonal low flow stages and nutrient-rich fine sands may
support the growth of annual or perennial herbs and shrubs on mid-channel and transverse
bars in braided rivers, this sparse vegetation cover has negligible impact upon sediment
deposition and erosion patterns, and is removed easily at flood stage (Coulthard, 2005). This
mutual interaction between vegetation and erosion-deposition can be viewed as a threshold
condition: if sufficient vegetation establishment occurs, resistance may exceed the erosion-
deposition capability of a normal flood such that stabilization ensues, prospectively altering
sedimentation patterns, increasing bank strength, and reducing channel width-depth ratio
(Gran and Paola, 2001; Coulthard, 2005; Eaton et al. 2010). In anabranching channels the
vegetation cover on mid-channel bars inhibits lateral migration, inducing a stable branching
channel condition. During lower frequency floods, when bars are partially or completely
submerged by flow, vegetation increases flow resistance, traps sediment, and inhibits erosion.
This study builds upon previously-reported exploratory analyses of river diversity in the
source zone of the Yellow River (Blue et al., 2013; Brierley and Huang, 2013; Li et al., 2013;
Yu et al., 2014). In this region, herbs and sparse shrubs that establish on the sand/gravel bars
of braided rivers have a trivial influence upon channel morphodynamics, while establishment
of dense shrubs and sparse trees on sand/gravel bars promotes the emergence of anabranching
channel configurations. Building on these observations, a vegetative gradient of river
morphologic adjustments is established for four reaches: Dari and Maqu reaches of the
Yellow River main stream, and Daheba and Lanmucuo River tributaries of the Upper Yellow
River (Table 1). Dari reach has a semi-stable braided channel, where sandbars are covered by
herbaceous vegetation and sparse shrubs. Maqu reach has a very stable anabranching channel
with dense willows (*Salix atopantha*) on sandbars. The study reach along Lanmucuo River
has a stable gravel meandering river with herb coverage. The study reach along Daheba River
has a very unstable gravel braided channel without vegetation cover. We develop and apply a
simplified model to explain the interaction of sediment transport capacity and river bed
deposition in these reaches, examining the effect of vegetation resistance and adjustment of





fluvial hydraulic geometry. From this, we quantitatively analyse the stability and evolution of
braided, anabranching, and meandering reaches during flood events.
**2    Study area and methods**
Upstream of Tangnaihai hydrological station the source zone of the Yellow River drains an
area of 122,000 km$^2$ (see Fig.1).  In the 1950s the Yellow Water Conservancy Commission
established four hydrological stations in this area, namely (from upstream to downstream),
Huangheyan station in Maduo County, Jimai station in Dari County, Maqu station in Maqu
County, and Tangnaihai station in Xinhai County. The reach from Huangheyan to Jimai
station is 325 km long and drains an area of 24,089 km$^2$. In this reach the valley is quite wide,
with semi-braided and semi-anabranching planform morphologies characterized by disordered
channels with a large number of bars. The reach from Jimai to Maqu is 585 km long and
drains an area of 41,029 km$^2$. The upper section of this reach has a deeply incised (confined),
sinuous valley between the Anyemaqen and Bayan Har Mountains. Flowing into the Ruoergai
alluvial basin, there is a diverse array of planform types (Blue et al., 2013; Li et al., 2013).
The reach from Maqu to Tangnaihai station is 373 km long and drains an area of 35,924 km$^2$.
Most of this reach comprises a steep and incised canyon.
For this study, field investigations of vegetative influences upon bed/bar geomorphic
processes were conducted four times in the summers of 2011-2014. Particle size distributions
of bed and bank materials size were analyzed using a laser particle size analyzer (Mastersizer
2000) and field sieves were used to test ten samples of river bed and bank materials in each
reach. As a supplement, photographs of gravel and cobbles on the bed/bar surface were taken
to visually estimate bed particle size. To estimate bedload transport capacity, water depth was
measured approximately in the field and channel width using remote sensing images of the
branching channel network. Remote sensing images from 2005-2014 were downloaded from
Google Earth (resolution of about 0.24 m).
The best available hydrological data that could be accessed for this study were daily stage-
discharge data from Jimai (1964-1985), monthly stage-discharge data from Maqu (1959-
1970), monthly cross-section elevation change data from Shangcun station along the Daheba
River (1.8 km upstream from its confluence with the Yellow River; 2009-2011), and 2011-
2014 field data for the Lanmucuo River (a tributary of the Yellow River in Maqu-Tangnaihai
section, at an elevation of 3400-4200 m a.s.l., for which upstream and mid-catchment reaches
have a typically meandering channel, while the downstream reach has a confined bedrock





channel). There are no intensive human activities in this area of the Yellow River Source
Zone.
**3    Basic characteristics of four alluvial reaches**
Fig.2 shows the planform morphology of the four channel reaches, and Figure 3-6 show bars,
the channel bed, and bed sediment. Basic channel characteristics of the study reaches are
summarized in Table 2.
Dari reach has a semi-braided and semi-anabranching channel in a wide valley (Fig.2 (R1)
and Fig. 3). This braided-anabranching transition zone is considered to be semi-stable, with
an active channel zone that is around 1 km wide. The braided part of the channel is made up
of many small mid-channel and transverse bars, with multiple connected branching channels.
In the anabranching part, the large bars/islands are covered by dense grassland vegetation.
Given the extensive width of the active channel zone, annual floods during June-September
exert negligible impacts upon these relatively stable surfaces.
The Maqu reach is located in a wide alluvial valley. The dense tree cover of the vegetated
islands is indicative of a stable channel configuration (see Fig. 2(R2) and Fig. 4). During the
flood season, tree trunks are partly submerged into water, but the dense trees are sufficiently
strong to limit bed erosion. As a result, the anabranching system as a whole is quite stable
with high bedload transport capacity.
Lanmucuo River is a meadow meandering river with nearly 100% vegetation cover(see Fig.
2(R3) and Fig. 5). The root system of riparian grasses induces considerable protection from
near-bank erosion. Field investigation from 2011-2014 indicate that the lateral migration
induced by cantilever bank failure occurred at a rate less than 0.2 m/yr. The gravel-bed
channel has a low bedload transport rate in the flood season. In some local sections, mid-
channel bars with dense grass coverage have developed at the apex of bends. The whole
channel is quite stable, in spite of short-term outer bank failures and long-term meander neck
cutoffs.
Daheba River has incised into the Gonghe-Xinhai sedimentary basin. Severe gully erosion has
incised river-lacustrine sediments to a depth of 50-100 m, supplying large volumes of
gravel/cobble to the middle and lower Daheba channel. Excessive sediment supply has
resulted in continuous aggradation of the channel bed along middle and lower courses of the
Daheba River. Alluvial fans in gully outlets not only supply additional sediment, but also
push the channel to the opposite side of the valley floor (a big fan is shown near D point in





Fig. 2(R4) and Fig. 6). As a result, the main branching channels are subjected to frequent and
recurrent avulsion. Flows erode new small branching channels during the flood season, but a
main channel coexists with several branching channels in the non-flood season. Unstable mid-
channel bars are unvegetated other than sparse vegetation coverage (grass and shrubs) on
riparian banks. The gravel-cobble bed and high bedload transport rate restrict vegetation
establishment and growth, resulting in a typically unstable braided river.
Bank strength induced by sediment material mix and vegetation root networks exerts a critical
influence upon the stability of alluvial channels (Eaton and Giles, 2009). Reinforcement of
bank strength reinforced by grass, shrub, and tree roots is related to the density, depth, and
spatial structure of the root network (Abernethy and Rutherfurd, 2001). Fig.7 shows
representative photographs of river banks in the four study reaches. The diverse bank material
composition and vegetation cover affect the relative strength of banks and their capacity to
resist nearbank flow scour. The river bank in Dari reach has a two-layer structure, with a 20-
30 cm deep soil-root layer ($d_{50} = 0.02$ mm) lying atop a gravel-sand layer ($d_{50} = 6.0$ mm) (Fig.
7(a)). The river bank in Maqu reach has a dense grass, shrub, and tree cover (Figure 7 (b)),
with no indication of flow scour in the flood season. The study reach along Lanmucuo River
has a typical composite bank sedimentology of a mixed load river (Fig. 7(c)). An upward-
fining sequence is characterized by a basal gravel unit ($d_{50} = 5.5$ mm) extending to a 10-30 cm
thick silt/sand layer ($d_{50} = 0.03$ mm) that is capped by a 10-50 cm thick fine-grained soil-root
complex ($d_{50} = 0.02$ mm). Conversely, the bank of the middle Daheba River has characteristic
deposits of a bedload-dominated river, with gravel and a sparse grass cover (Fig.7(d)).
Adjacent terraces that are more than 10m high limit the capacity for channel widening, while
actively supplying gravels. Mobile gravel banks influence the braided characteristics of
Daheba River. In summary, bank strength of the four study reaches varies from high to low as
follows: Maqu reach, Lanmucuo River, Dari reach, and Daheba River.
**4   Estimation of bedload transport capacity**
Given the lack of observed data of bed load transport rate, bedload transport capacity has been
estimated for a rectangular cross-section using the theoretical bed load formulae outlined
below. Channel flow follows the laws of flow continuity, flow resistance and sediment
transport with flow continuity law taking the form:





$Q = AV$ .                                                                     (1)
where $Q$, $A$, and $V$ are flow discharge, channel cross-sectional area, and average flow velocity,
respectively, $A=WH$, $W$ is channel width, $H$ is water depth.
Field observations shows that anabranching rivers on the Northern Plains of arid central and
northern Australia flow over largely plane beds (e.g. Tooth and Nanson, 1999, 2000; Tooth,
2000; Jansen and Nanson, 2004), so this study adopts the Manning formula to embody the
law of flow resistance for uniform alluvial channel flow:
$V = \dfrac{1}{n} R^{2/3} S^{1/2}$ .                                           (2)
where $R$ is hydraulic radius, $R=WH/(2H+W)$, $S$ is flow energy slope, $n$ is Manning's
roughness coefficient. In this study, following Chow (1959), $n = 0.050$ if no vegetation in
gravel-bed channels at high stages, $n = 0.030$ in floodplain with short grass, $n = 0.050$ in
floodplain with scattered brush and heavy weeds, and $n = 0.150$ in floodplain with dense
willows at flood stage.
Bedload transport fashions channel form and evolution for these gravel-bed rivers. Among
numerous bedload formulae, the Meyer-Peter and Muller equation has been extensively and
successfully applied (Meyer-Peter and Müller, 1948). The modification developed by Wong
and Parker (2006) has been used in this study:
$\Phi = 3.97(\Psi - 0.0495)^{3/2}$ .                                            (3)
where $\Phi$ and $\Psi$ are the dimensionless bedload transport rate per unit channel width and the
dimensionless flow shear stress, respectively, that are defined as
$\Phi = \dfrac{q_b}{\sqrt{(\rho_s / \rho - 1) g d^3}}$ .                         (4)
$\Psi = \dfrac{RS}{(\rho_s / \rho - 1) d}$ .                                     (5)
where $q_b$ is the dimensional bedload transport rate per unit channel width, $\rho_s$ is the density of
sediments transported, $\rho$ is the density of water, $g$ is the acceleration of gravity, $d$ is the
median sediment size.





Cross-section and water depth were measured based on field survey and remote sensing
images (see Table 2). Estimated hydraulic parameters and bedload transport capacity for the
four reaches, derived using Eq.(1)-(5), are summarised in Table 3. Note that channel width is
effective bankfull width in the flood season, not valley width due to the existence of bars. The
adopted mean grain size is lower than bed sediment size.  These results are considered to be
approximations, at best, and are analysed here solely in relational rather than absolute terms.
Results show that the bedload transport capacity of the four reaches from high to low is as
follows: Maqu, Lanmucuo, Dari, and Daheba reaches.
**5   Effect of vegetation and bedload capacity on channel stability**
**5.1.1  Dari reach (braided/anabranching river with grass and shrub cover)**
Dari reach is a wide semi-braided and semi-anabranching channel, where the channel width is
up to 1600 m (Fig.3(a)). Some large stable gravel bar or islands have a dense grass and sparse
shrub cover. Many unstable bars with low vegetation cover are subjected to recurring erosion
and channel adjustment. Vegetation may inhibit erosion and enhance bar stability at middle
flood stage, but the resistance effect of vegetation at high flood stage is very limited.  As a
result, the whole channel may be eroded at high flow stage, resulting in disordered patterns of
mid-channel gravel bars.  The estimated bedload transport capacity per unit channel width is
1.77 kg/m/s for 2.0 m water depth (see Table 2). If the water depth increases to 3.0 m in the
flood season, bedload transport capacity per unit width significantly increases up to 14.93
kg/m/s. It is likely that these flow depths cause intense erosion that divides the stable bars into
many unstable bars.
Fig. 8(a) and (b) show monthly stage-discharge relationships for 1968 and 1984, respectively.
Since Dari reach is a multi-thread channel system, the stage-discharge relationship is not a
single function relationship. In non-flood months (December, January, February, March, and
April) the river bed is frozen. May and November are pivotal times in the stage-discharge
relationship (the former reflects ice melt, the latter freezing). In flood months (June, July,
August, and September) the stage-discharge relationship adjusts due to strongly erosion and
deposition within the channel.  For instance, different discharges for the same flow stage in
June and July 1968 are considered to reflect erosion of the channel (Fig. 8(a)). In the other
instance shown here, the maximum discharge in 1984 occurred in July (Fig.8(b)), probably
marking the transition from erosion to deposition phases.





Fig.9 shows the stage-discharge relationships of the Upper Yellow River at Dari from June to
September in 1964-1984.  Apparently, the stages of 1975 are out of line with 1978, perhaps
indicating that the elevation benchmark of the station occurred in 1976 or 1977. In the same
month of different years, the stage-discharge relationship does not have a simple
corresponding relation, especially in August and September. This may reflect: 1) responses of
the channel bed to strong deposition in June and July, and thereafter the high stage
corresponds to low discharge such as August in 1978-1984 and September in 1964-1975; 2)
the channel bed strongly erodes in June and July, and thereafter the high stage corresponds to
high discharge such as August in 1964-1975 and September in 1978-1984. Overall, Figures 7
and 8 indicate that the channel of Dari reach is quite unstable during the flood season, with
erosion and deposition changing the stage-discharge relationship. A sketch showing how flow
erosion divides bars and deposits to form new bars is shown in Fig. 10.

### 5.1.2  Maqu reach (anabranching river with tree cover)

Maqu reach in wide Ruoergai basin is covered by dense tress (*Salix atopantha*) and has a
stable anabranching channel planform (Fig. 4a).  It is postulated that a herb and shrub cover
gradually supports the stabilization of new bars, facilitating sediment deposition on the body
of the bar during low and middle flood stages, and protecting the bar from erosion at high
flood stages. Subsequent development of trees presents a tall green barrier in the flood period
(Fig. 12). Although the water floods trees, their density induces sufficient resistance to
decrease the flow velocity and trap fine sand and gravel on the body of the bar. Therefore, this
anabranching channel system is basically stable over a decadal timescale.
Water stage change at Maqu station from 1959-1970 is shown in Fig. 11. The stage peak
occurs in July and September. The maximum difference of 2.43 m occurred between June and
September in 1963. If the water depth increases to 8.0 m from 5.5 m, bedload transport
capacity increases to 18.52 kg/s/m from 7.63 kg/s/m. As a result, the branching channel bed
may erode if upstream sediment supply exceeds the transport capacity. However, protection
by trees is strong enough to inhibit erosion of bars.  In contract, if the transport capacity
surpasses the upstream sediment supply, increasing bed deposition with flow stage further
increases the transport capacity of the reach. This agrees with analyses by Huang and Nanson
(2007) who stated that anabranching channels can achieve the optimal transport efficient
without increasing bed gradient. Even though these reaches may appear to promote deposition
on the channel bed during extreme floods (see Fig. 12), the flow erodes the bed later in the





flood season, thereby maintaining an equilibrium cross-section. As a result, the anabranching
channel of Maqu reach maintains a long-term stable situation.

### 5.1.3 Lanmucuo River (passive meandering river with meadow cover)

Lanmucuo River is a typical meandering river covered by dense meadow. Although typically
characterized by large bends in a flat valley, mid-channel gravel bar covered by herbs
sometimes form at the apex of bends (Fig. 5(a)). The meandering channel and bars are very
stable because of low sediment supply in the flood season and good vegetation coverage. The
tight root-soil complex on concave banks inhibits flow scour. When cantilevered bank failures
do occur, slump blocks restrict further erosion of the bank. Grass develops on the point bars
of convex banks. When the overbank flow submerges the point bar, the herbaceous vegetation
can increase flow resistance and promote fine sand deposition (Fig.13), thereby maintaining
channel geometry with a relatively low migration rate. Growth of herbs on mid-channel bars
an apices (Fig.5(a)) helps to increase the flow resistance and trap fine sediment, facilitating
channel stability.

### 5.1.4 Daheba River (unvegetated braided river)

The gravel bed of Daheba River is characterized by deposition in the flood season and erosion
in the non-flood season. This makes it difficult for vegetation to develop on bars and banks of
the braided channels. Fig.14 shows morphological changes of the riverbed before and after
the flood season in 2005. The main branching and sub-branching channels of the channel
completely changed, with an initial phase of sediment deposition followed by flood-induced
division of bars and the re-emergence of a multi-thread braided system. Table 3 shows
derived estimates of the bed load transport capacity per width, $q_b$=0.47 kg/s/m. This capacity
is seemingly unable to efficiently transport the excess sediment supply from upstream. As a
result, serious deposition occurs along Daheba River in the flood season.
Adjustments to channel geometry as a result of erosion and deposition processes before,
during and after the flood season are shown in Fig.15. The elevation of the riverbed on July
29 2009 was 0.27 m higher than on April 1 2009. Other than slight erosion of the left bank,
the subsequent phase was depositional, with up to 1.59 m of aggradation occurring by
October 23 2009. The elevation of riverbed was increased by 0.27 m after the flood season in
2010.  The elevation of the riverbed in July 1 2011 was 0.26 m higher than on April 29 2011.
Trivial deposition occurred from July 1 to July 8, but 0.24 m of erosion occurred by July 23,





with subsequent deposition of 0.27 m by October 23. As a result, the riverbed elevation was
0.24 m higher after flood season in 2011, but multiple phases of deposition and erosion has
occurred. The deposition-erosion-deposition phases may reflect lower bedload transport
capacity relative to sediment supply in the early flood season, but widespread deposition
increases the local bed slope, thereby increasing bedload transport capacity. According to Eq.
(3), a 10% increase in bed slope increases the transport capacity by 85% in Daheba reach, so
bed erosion occurs again. Bed erosion decreases the bed slope until the transport capacity has
adjusted to reduced sediment supply, thereby inducing riverbed deposition once more.
Consequently, alterative deposition and erosion leads to the extreme instability in the middle
and lower Daheba River.
**6    Discussion and Conclusions**
This study has outlined the complex interplay between vegetation and bedload transport and
channel stability in four reaches of the Yellow River source zone. Bar morphodynamics are
shown to exert a key control upon the behaviour of braided, anabranching, and meandering
channels (Hooke, 1986; Kleinhans, 2010; Kleinhans and van den Berg, 2010; Church and
Ferguson, 2015). Bar development and stability reflect the ability of vegetation to trap
sediments and stabilize banks, which in turn is directly influenced by flow energy
relationships (i.e. these are mutual adjustments; Corenblit et al., 2007; Gurnell et al., 2012;
Gurnell, 2014; Osterkamp and Hupp, 2010; Pietsch and Nanson, 2011). In this study, riparian
vegetation and its root network are considered to restrict channel width and increase hydraulic
efficiency, inducing greater bedload transport capacity in multi-thread channels
(Allmendinger et al., 2005; Huang and Nanson, 2007). Islands and floodplains are able to trap
more fine-grained sediment in the flood season, enhancing the longer-term (decadal) stability
of anabranching channels, as shown by the stable islands of Maqu reach.
Relative to the passive (resisting) role of vegetation, bedload transport actively affects short-
term patterns and rates of bed erosion and deposition. This, in turn, is affected by
relationships between the flow regime (especially flood events and formative flows) and the
influence of sediment supply upon bedload transport for differing river types (Church and
Ferguson, 2015; Dunne et al., 2010). The supply of bed material sediment to an alluvial
channel accelerates the growth of mid-channel, transverse, and point bars, thereby enhancing
thalweg development and locally increasing flow velocity. Non-equilibrium between
sediment supply and transport induces local channel instability, accentuating either bed



erosion or deposition (Jansen and Nanson, 2010; Nanson and Huang, 2008). In this study, a
channel stability gradient accords with both sediment movement and vegetation cover,
wherein bedload transport capacity (a function of bed slope, hydraulic geometry, and
sediment particle size) is related to the influence of riparian vegetation upon channel
geometry/planform.
In summary, channel stability of four alluvial reaches in the Yellow River source zone reflects
interactions between channel geometry/planform, bedload transport capacity, sediment supply
in the flood season, and the geomorphic/hydrodynamic role of vegetation cover on the valley
floor. Although the elevation of four reaches is different (Dari = 3960 m, Maqu = 3465 m,
Lanmucuo River = 3604 m, and Daheba = 2832 m), the precipitation, temperature, and bed
sediment size are basically similar (Yu et al., 2014). Nevertheless, vegetation coverage in the
four reaches is quite different. The Dari reach (anabranching-braided) has a herb and shrub
cover, Maqu (anabranching) reach has trees, Lanmucuo River (meandering) has meadow, and
Daheba River (braided) has no vegetation cover. We contend that the differing vegetation
cover and planform response reflects the delicate balance between erosion and deposition on
the channel bed and bank as influenced by bedload sediment supply in the flood season. Only
when the bedload transport capacity is equivalent or greater than sediment supply, does
vegetation act as a key determinant of channel stability.
**Acknowledgements**
This study is supported by Natural Science Foundation of China (NSFC,
Ref.41571009; 41330751; 51179089). Field assistance by X.Z. Wang, C.D. Zhang, and X.D.
Zhou (2011-2014) is greatly appreciated. Z.W. Li and G.A. Yu designed and conducted the
field investigation. G. Brierley and Z.Y. Wang supervised the research and discussed the
details. Z.W. Li prepared the manuscript with contributions from all co-authors.





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



1    Table 1. Characteristics of the four study reaches (Flood season = June-September)

| Alluvial reach | Planform type | Catchment area (km$^2$) | Flood-season mean discharge (m$^3$/s) | Channel gradient | Vegetation cover |
|---|---|---|---|---|---|
| Dari | braided-anabranching | 45020 | 270 | 0.00120 | dense grasses/ sparse brush |
| Maqu | anabranching | 86000 | 920 | 0.00050 | dense trees |
| Lanmucuo | meandering | 660 | 15 | 0.00150 | dense grass |
| Daheba | braided | 5200 | 70 | 0.00144 | non-vegetation |



1    Table 2. Characteristics and bed material of alluvial channels in the four study reaches

| Alluvial reach | Channel width (m) | Water depth (m) | Bed material $d_{50}$ (m) | Branching channels | Stability |
|---|---|---|---|---|---|
| Dari | 450-1600 | 1.0-3.0 | 0.025 | >5 | semi-stable |
| Maqu | 300-1000 | 2.0-5.0 | 0.015 | >3 | very stable |
| Lanmucuo | 10-20 | 0.3-1.0 | 0.030 | <=2 | very stable |
| Daheba | 150-500 | 0.5-2.0 | 0.060 | >3 | unstable |



1  Table 3. Estimation of hydraulic coefficients and bedload transport rates

| River reach | Bankfull channel width (m) | Bankfull water depth (m) | Channel gradient | Mean grain size (m) | Manning coefficient | Average velocity (m/s) | Channel discharge (m³/s) | $q_b$ (kg/s/m) |
|---|---|---|---|---|---|---|---|---|
| Dari | 200 | 2.0 | 0.00120 | 0.015 | 0.05 | 0.90 | 269.67 | 1.77 |
| Maqu | 400 | 5.5 | 0.00050 | 0.015 | 0.15 | 0.46 | 1003.55 | 7.63 |
| Lanmucuo | 20 | 0.8 | 0.00150 | 0.010 | 0.03 | 1.06 | 16.91 | 2.35 |
| Daheba | 50 | 1.5 | 0.00144 | 0.020 | 0.05 | 0.96 | 71.75 | 0.47 |





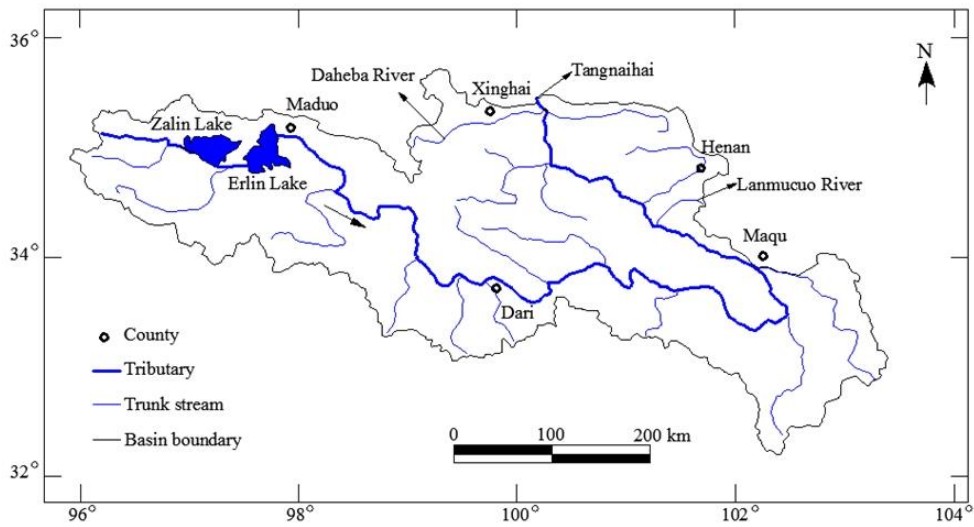

2    Figure 1. The course of the Upper Yellow River. R1 is Dari reach, R2 is Maqu reach, R3 is

3    Lanmucuo River, and R4 is Daheba River.



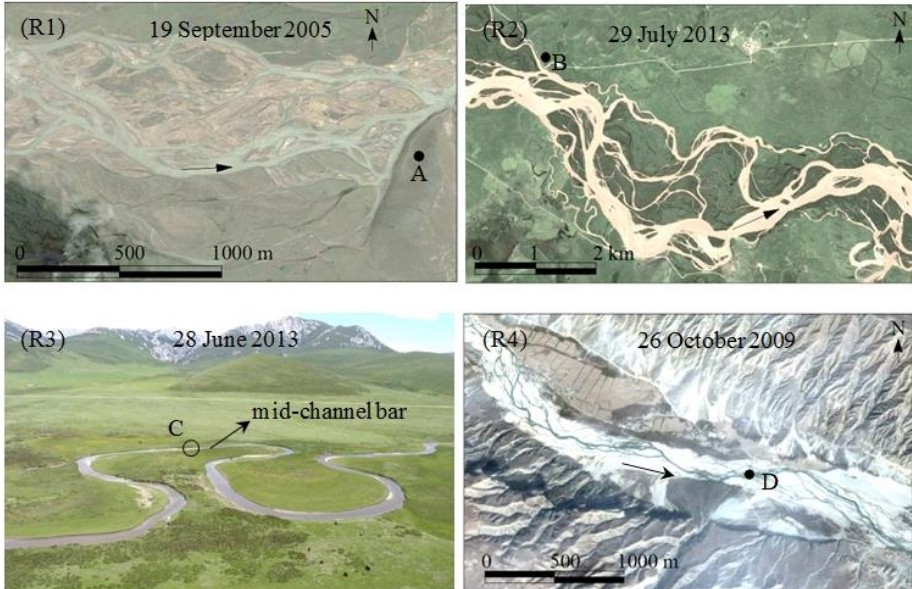

Figure 2. Planform morphology of the study reaches (R1 is Dari reach , R2 is Maqu reach, R3
is Lanmucuo River reach, and R4 is Daheba River reach). R1, R2, and R4 are Google Earth
images and R3 is a photograph taken from nearby hills. Points A, B, C, and D are the location
of photographs shown in Figures 3-6.





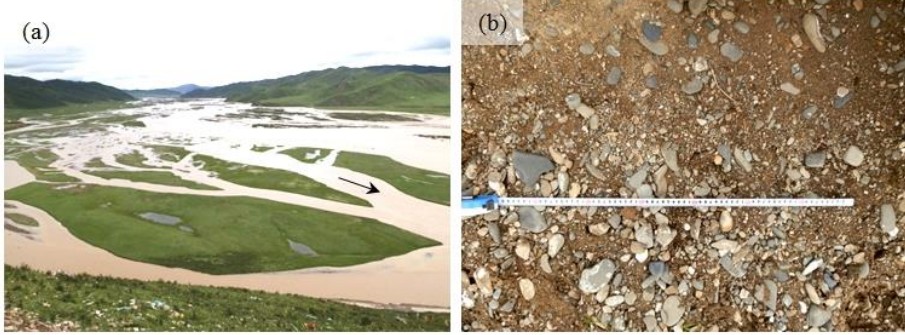

2    Figure 3. Channel morphology and gravel bed of Dari reach (photographs taken on 2 July,

3    2012, 33.7553 °N, 99.6414 °E, 3960 m elevation).




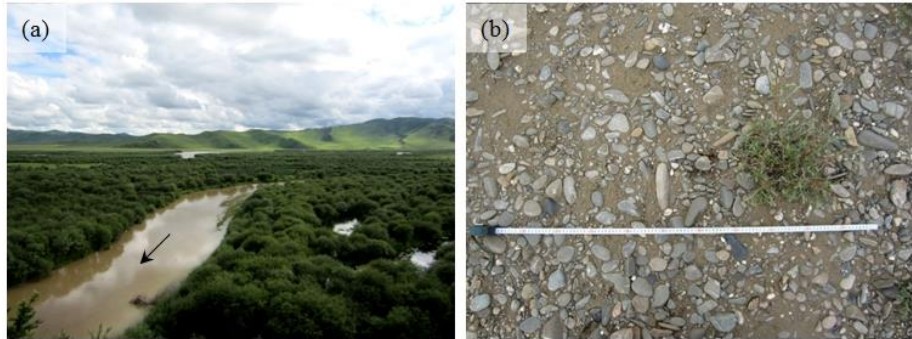

2    Figure 4. Channel morphology and gravel bed of Maqu reach (photographs taken on 8 July,

3    2012, 33.3594 °N, 102.0553 °E, 3465 m elevation).




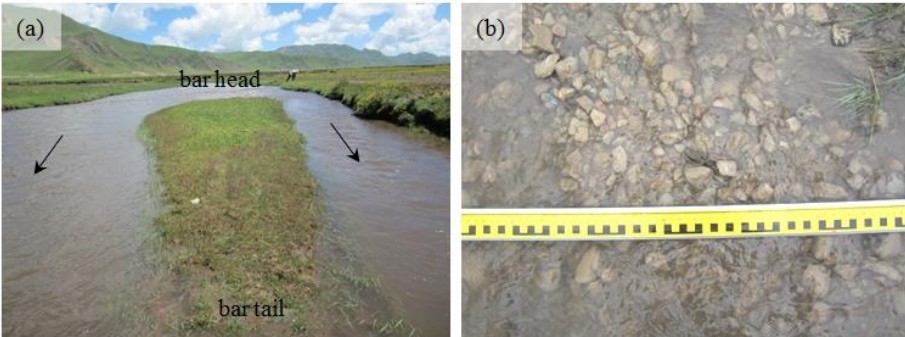

2    Figure 5. Channel morphology and gravel bed of a grass covered bar in middle Lanmucuo

3    River (to photographs taken on 5 July, 2012, 34.4287 N, 101.4663 E, 3604 m elevation).



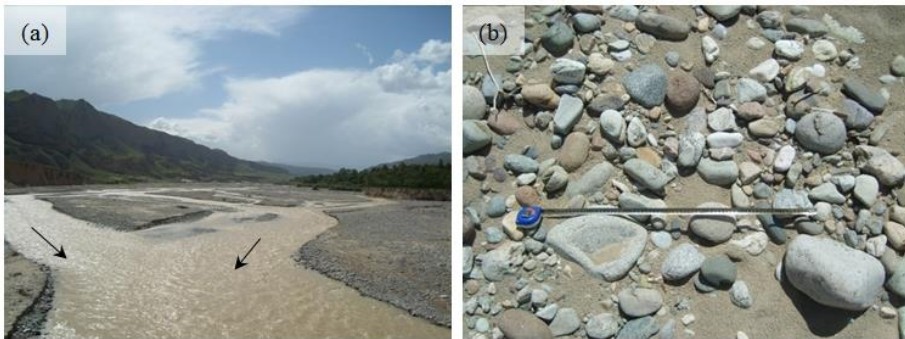

2 Figure 6. Channel morphology and gravel bed of middle Daheba River (photographs taken on

3 6 August, 2011, 35.5169 °N, 100.0183 °E, 2832 m elevation).





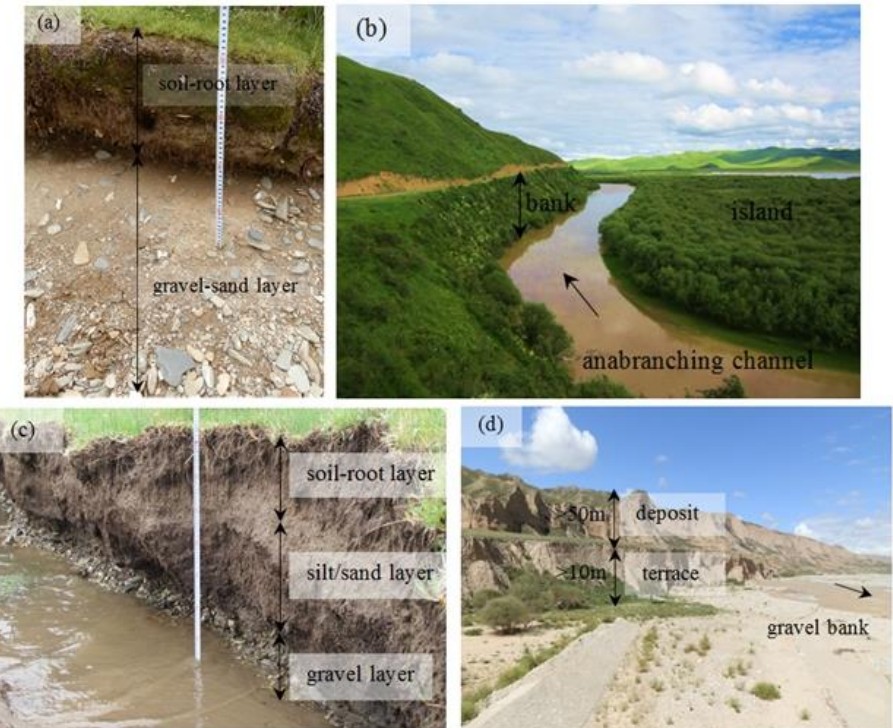

2    Figure 7. River bank of the study reaches (a) Dari reach, (b) Maqu reach, (c) Lanmucuo River

3    reach, and (d) Daheba River reach.



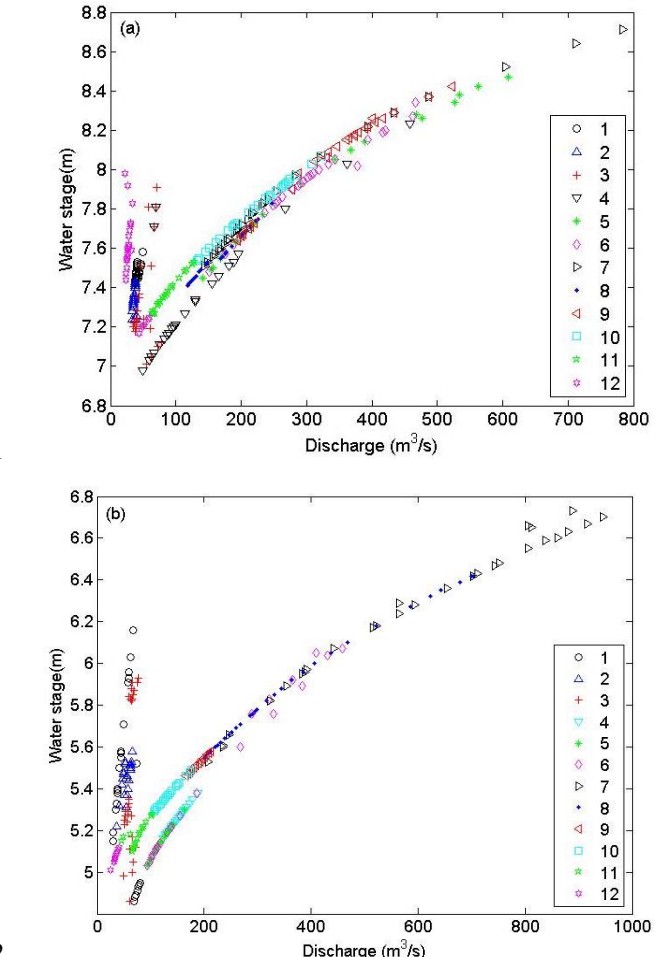

3    Figure 8. Monthly stage-discharge relationships for Jimai hydrological station in Dari reach (a)

4    1968 (b) 1984 (Note: number refers to month, e.g. 1 for January and 12 for December) .





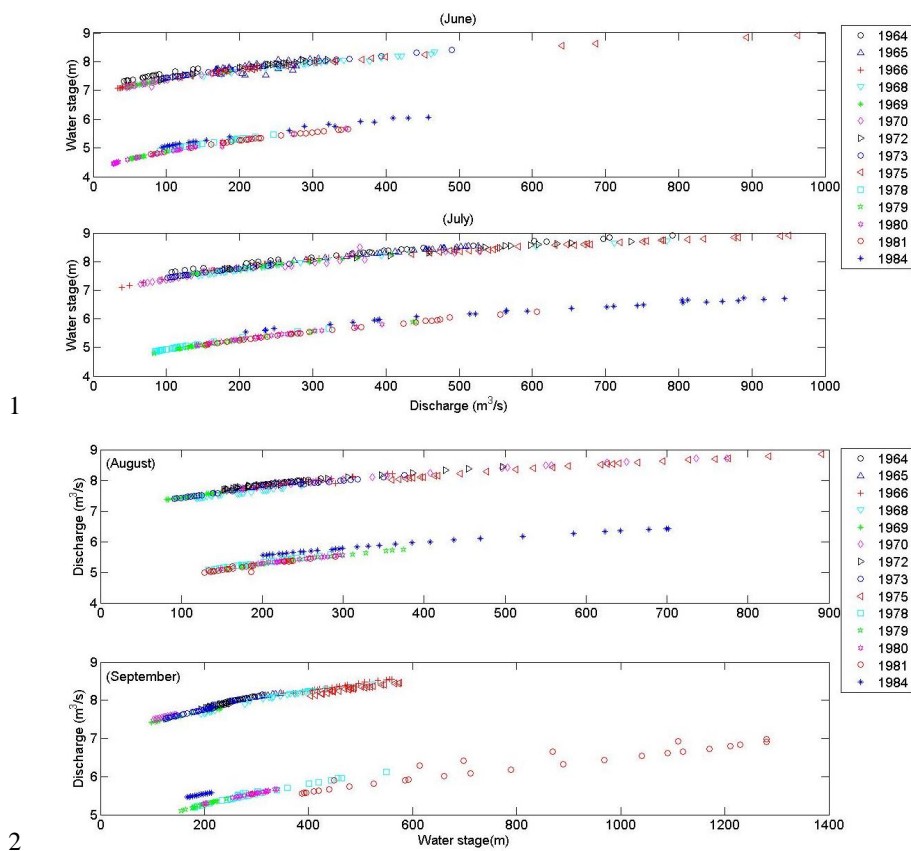

3    Figure 9. Annual stage-discharge relationship (1964-1984) of Dari reach in Jimai hydrological

4    station.



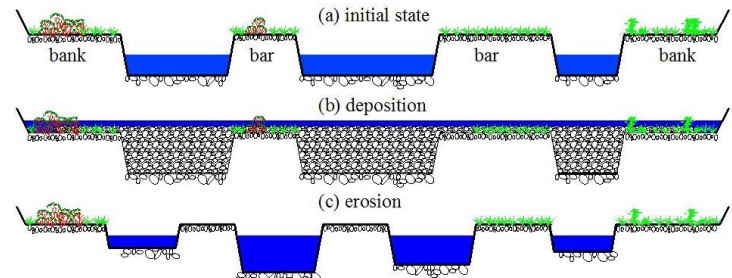

2    Figure 10. Sketch of channel bed deposition and erosion in flood season in Dari reach.





1  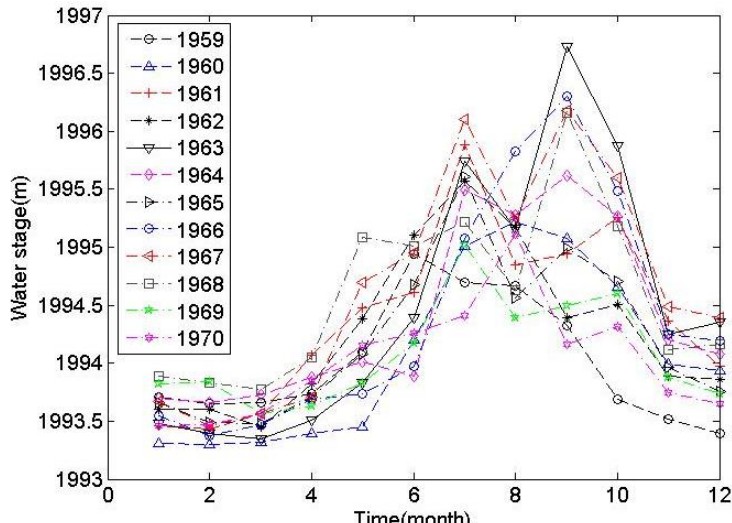

2  Figure 11. Monthly stage change of Maqu hydrological station (1959-1970).





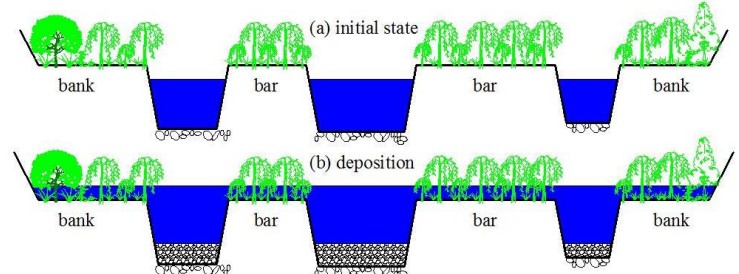

2   Figure 12. Sketch of branching channel deposition and stage increasing in flood season in

3   Maqu reach





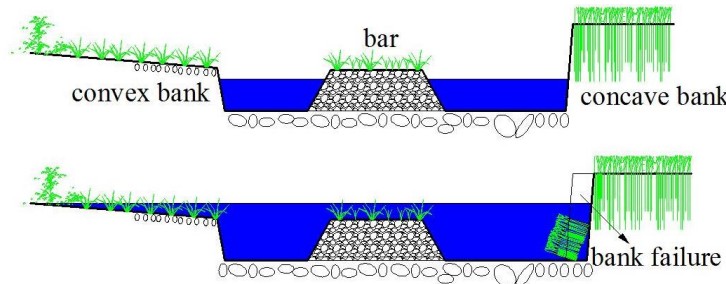

2    Figure 13. Sketch of submerged bend apex with a mid-channel bar in Lanmucuo River





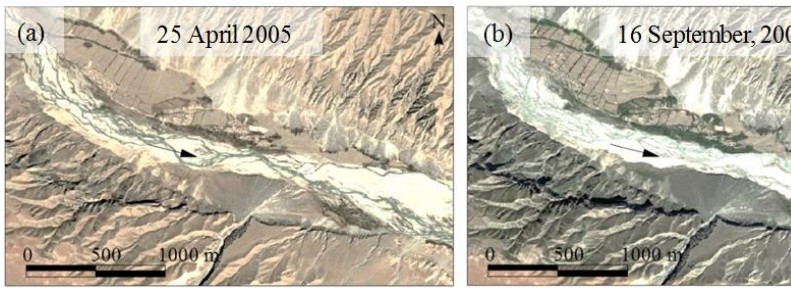

2    Figure 14. Braided channels evolution of the middle Daheba River in 2005 (a) in non-flood

3    season, (b) in flood season





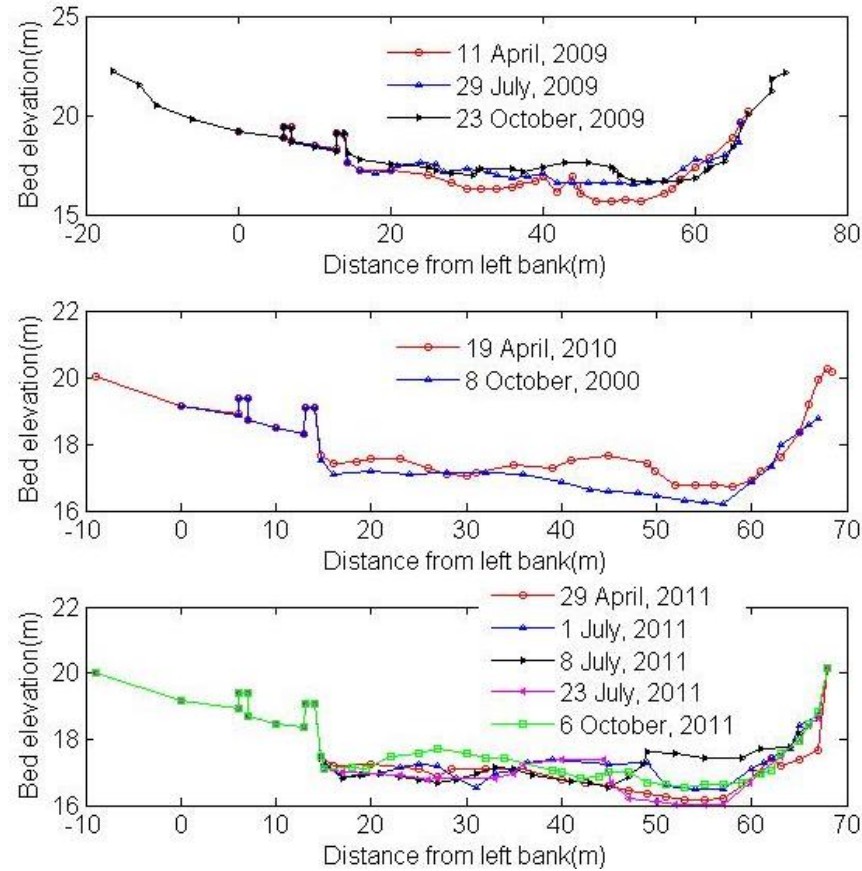

2    Figure 15. Elevation change of cross-section in Shangcun hydrological station (2009-2011)

3    (left for left bank, right for right bank)