# Peer review of "Vegetative impacts upon bedload transport capacity and channel stability for differing alluvial planforms in the Yellow River Source Zone"

_Hydrology and Earth System Sciences, 2015_

## Referee Comment (RC1) · P. Gao (Referee) · 22 Feb 2016

In this study, the authors attempted to explain the different channel plainforms of four reaches in the source area of Yellow River, China using (partially) measured water discharge, stage, and cross section data, as well as qualitative description. First, I think this area is unique regarding to the world large rivers and thus is worth studying. Second, the river dynamics in this area is very complex and hence is very hard to capture. Therefore, I think this study is significant and potentially very useful for understanding the river environment in source areas for large rivers in general. However, I think the authors need to fix a series of problems in the current manuscript before it reaches the level for publication. I describe them in details: Introduction: the authors spend

too many sentences describe the progress in channel plainforms, in particular braided and anabranching rivers (pages 2 to 5). Instead, I think they should reduce these part. In the meantime, they should expand the studies on river diversity in the source area of Yellow River (lines 17-31, page 5) and explain what we need to explore further in details, which could lead to the objectives of this study. There are quite a few English problems in this section. Sections 2 and 3: these two should be combined into one section. Also, Figs 1 and 2 should be combined. Section 4: it is very important that the authors specify what value of Manning's n are used for each of the four reaches when they introduce their model because comparison of their difference would provide a quantitative means of showing the impact of vegetation on river morphology. The sentence in lines 4-5 on page 9 does not make sense. Section 5.1.1: Page 10, Lines 14-15: what is the difference between middle and high flood stages? It might be more informative if this description is tied to Figure 8. For example, can one say that middle flood stage may be represented by the high discharges in September and high flood stage may be reflected by the high discharges in July? Page 10, Line 18: if a water depth of 2.0 m represents the bankful discharge, then what does the water depth of 3.0 m represent? Can I say that h = 2.0m is the height at the top of the stable bars in middle channels? Page 10, lines 23-32: the message delivered by this paragraph is very vague. It seems to me that the data in October in both 1968 and 1984 follow the curve formed by the data in June and July. If the authors believe there are significant difference between June and July, and August and September, why not use the data in the two periods to run non-linear regression (power function) and see if the exponents of the two are significantly different? The authors should explain quantitatively the geomorphological significance of the two different trends in Fig. 8a and 8b (i.e., the trend formed by the data in March and April against the trend formed by the remaining data). Also, the difference between the high scatter trend for low discharges (probably low flood stages) and regular trends for high discharges (probably high flood stages) should be elaborated. The key is to explain why channels in this reach is semi-braided and semi-anabranching. My guess is vegetation on bars assures that during

low and middle flood stages, bars and islands are relatively stable, while during high flood stage, they are unstable. Figure 8 should be used to make this point clear. Page 11, Lines 1-11: this paragraph is about Fig. 9. I think the figure shows a completely different aspect of stream channels in this reach: channel morphology before 1976 is different from that after 1976. This difference is represented by the two different trends of the data. The authors should run non-linear regression to establish power functions for the two different trends in each listed month and then link this difference to the possible difference of vegetation cover in the two different time periods. This would strengthen the analysis a lot. Fig. 10 is not well tied to the data shown in Figs. 8 and 9. It is nice, but there lacks evidence to support it. Section 5.1.2: First, the authors should mentioned Fig. 11 first and then Fig. 12. Second, the big problem here is that the postulation raised here (lines 15-19 on page 11) is not fully supported by the only data shown in Fig. 11. The stage data in Fig. 11 are not sufficient to argue the change of flow regime exactly because the channels here are anabranching channels. This means that the same flow stage in different seasons might be associated with different water discharges. Maybe there are no water discharge data available in this reach. If this is the case, the authors should re-think their arguments: the fact that these channles are stable means that sediment (bedload) supplied from upstream (i.e., the Dari reach) is balanced by the sediment transport capacity in this reach. One way might be useful is to compare the supplied bed load based on the prediction made for the Dari reach with the transport capacity predicted in this reach. The authors should expect that they are similar or very close to each other. Then, the impact of vegetation on the hydraulics might be reflected in Manning's n used in the bedload model. Comparing this value with the one used in the Dari reach may show the impact of vegetation on the stable status of this reach. Section 5.1.3: This reach is a tributary. If the authors have no water discharge data in this tributary, I suggest to delete this part completely from the current manuscript. This is because only showing a postulated diagram (i.e. Fig. 13) is insufficient to convince the readers about the status of this reach. Section 5.1.4: This reach is unstable. Again, just using the temporal changes of channel cross

sections between three years (i.e., Fig. 15) is not enough to explain how vegetation affects them. Again, I think it is very important for the authors to predict bedload transport rates and then use them to calculate the mean sediment load in this reach. By comparing this (or these) mean value(s), the authors may argue that why the reach is not stable. In the meantime, comparing the value of manning;s n used in this reach with those used in the first and second reaches along the main river would provide evidence of the impact of vegetation on river morphology. Minor points: Lines 5-8 on page 6: this description is very confusing; Lines 17-18 on page 7: why should the stable reach have high bedload transport capacity? Lines 4-5 on page 9: what does the rivers in an arid area have anything to do with rivers in the study area? Figure 1: Please mark R1, R2, R3, and R4. Also, only use the arrow to show the direction of flow. In the legend, 'Tributary' and 'Trunk stream' should be reversed. Please use 'Main stream' rather than 'Trunk stream'; Figures 3-6: these figures should be combined into one figure; Figure 8: please use the same legend for the two figures; Figure 14: it does not help much in understanding the difference between the regular and flood conditions;

---

## Referee Comment (RC2) · M. Coenders-Gerrits (Referee) · 23 Mar 2016

The authors present a relevant study on the effect of vegetation on bedload transport capacity and channel stability. Therefore, they study 4 reaches of the upper Yellow River, China. The 4 reaches differ in planform. Despite the potential interest, the paper is highly descriptive and hypothetical. Barely any data is collected to justify the conclusions. This leads to the question what we can learn from this study. The river planform is not really something we can easily adjust and the role of vegetation is more a result of the planform, than a cause. Maybe this also relates to the fact that there is no study objective given.

Abstract:
The abstract starts immediately with describing what the study entails, but the existing knowledge gap is missing. As well as the 'reason for this study'.

Introduction:
The introduction is really long and very general. It seems like a 'lecture' on river planforms in relation to bars. I would advise to shorten the introduction and focus on what is currently missing (knowledge gap) and why this study is relevant (what will it bring). Furthermore, I would also explain how the existing study differ from exiting studies.

P9L1-25:
add dimensions or units to symbols

Equation 3-5:
Why do you need eq 4 if you can also derive it from Eq 3 and 5?

Section 5:
Based on what can the authors conclude how the bars are developed/eroded? (fig 10, 12,13). Can this not better be answered with satellite images over several years?

Figure 1:
Naming R1, R2, R3, and R4 are not visible in the figure

Figure 8:
What's happening during the low flows? This seems to weird behaviour. How can the stage drop when Q increases? That is remains constant is possible if the river width increase after a certain threshold, but this seems unrealistic. Please elaborate/explain.

Figure 9:
Please be consistent. The upper graphs are Qh-plots, while the lower two are hQ-plots. Furthermore, the coloring is not that clear, which makes the plot difficult to interpret.

Figure 11:
Is the stage unit correct? What is the datum of this stage?

Throughout the entire manuscript:

- Textual : after "i.e." and "e.g." a comma should be placed

- Order appearance figures in text, is order figure numbers (e.g., figure 11 and 12). Please check

---

## Author Response (AR1)

**Response to Rereree #1**

**We would like to much thank Prof. Peng Gao (Referee #1) for his detailed comments**
**and suggestions on the original manuscript. These comments and suggestions have been**
**used to greatly improve the manuscript during the following revision. A point-by-point**
**response to your comments is addressed below.**

In this study, the authors attempted to explain the different channel planforms of four reaches
in the source area of Yellow River, China using (partially) measured water discharge, stage,
and cross section data, as well as qualitative description. First, I think this area is unique
regarding to the world large rivers and thus is worth studying. Second, the river dynamics in
this area is very complex and hence is very hard to capture. Therefore, I think this study is
significant and potentially very useful for understanding the river environment in source areas
for large rivers in general. However, I think the authors need to fix a series of problems in the
current manuscript before it reaches the level for publication.

**Response: Many thanks for your positive comments and pointing out the weakness**
**about the manuscript. We have updated the manuscript according to your suggestions.**

I describe them in details: Introduction: the authors spend too many sentences describe the
progress in channel planforms, in particular braided and anabranching rivers (pages 2 to 5).
Instead, I think they should reduce these part.

**Response: We have shorten the introduction and focus on the knowledge gap from the**
**existing studies. We feel it is important to contextualize this study, highlighting**
**similarities and differences with conventional literatures. We paid very careful attention**
**to this issue.**

In the meantime, they should expand the studies on river diversity in the source area of
Yellow River (lines 17-31, page 5) and explain what we need to explore further in details,
which could lead to the objectives of this study. There are quite a few English problems in
this section.

**Response: Good suggestions. We enhanced this part so as to make our studies more sense, emphasizing how and why this intriguing yet understudied part of the work relates to other areas. Meanwhile, the English has been polished and double-checked.**

Sections 2 and 3: these two should be combined into one section. Also, Figs 1 and 2 should be combined.

**Response: Section 2 and 3 have been combined into one section. Figs 1 and 2 have also been combined.**

Section 4: it is very important that the authors specify what value of Manning's n are used for each of the four reaches when they introduce their model because comparison of their difference would provide a quantitative means of showing the impact of vegetation on river morphology. The sentence in lines 4-5 on page 9 does not make sense.

**Response: We have specified Manning's $n$, and discuss variability in roughness. We reconsider the sentences.**

Section 5.1.1: Page 10, Lines 14-15: what is the difference between middle and high flood stages? It might be more informative if this description is tied to Figure 8. For example, can one say that middle flood stage may be represented by the high discharges in September and high flood stage may be reflected by the high discharges in July?

**Response: Yes, the difference between middle and high flood stages is not very distinct. Here the middle stage means that the channel flow partly submerges the bar surface, but the stage does not completely inundate the vegetation. Therefore, we need to add more explanation on the middle and high flood stages in the section.**

Page 10, Line 18: if a water depth of 2.0 m represents the bankfull discharge, then what does the water depth of 3.0 m represent? Can I say that h = 2.0 m is the height at the top of the stable bars in middle channels?

**Response: Sure. If a water depth of 2.0 m represents the bankfull discharge, the water**

**depth of 3.0m represent that 1m is inundation water depth. We can think $h$=2.0m is the**

**height at the top of the stable bars surface, but does not submerge the vegetation (i.e.,**

**trees)**

Page 10, lines 23-32: the message delivered by this paragraph is very vague. It seems to me that the data in October in both 1968 and 1984 follow the curve formed by the data in June and July. If the authors believe there are significant difference between June and July, and

August and September, why not use the data in the two periods to run non-linear regression (power function) and see if the exponents of the two are significantly different?

**Response: Using non-linear regression for the data in different months in both 1968 and**

**1984 is very good choice. We have done this job to obtain the exponents so as to**

**quantitatively explain the difference.**

The authors should explain quantitatively the geomorphological significance of the two different trends in Fig. 8a and 8b (i.e., the trend formed by the data in March and April against the trend formed by the remaining data). Also, the difference between the high scatter trend for low discharges (probably low flood stages) and regular trends for high discharges (probably high flood stages) should be elaborated. The key is to explain why channels in this reach is semi-braided and semi-anabranching. My guess is vegetation on bars assures that during low and middle flood stages, bars and islands are relatively stable, while during high flood stage, they are unstable. Figure 8 should be used to make this point clear.

**Response: We agree with this suggestion regarding analysis on Fig.8a and Fig. 8b and**

**their underlying meaning. Accordingly, we have explained quantitatively the**

**geomorphological significance of the two different trends and why channels in this reach**

**are semi-braided and semi-anabranching, further, emphasizing the role of vegetation.**

Page 11, Lines 1-11: this paragraph is about Fig. 9. I think the figure shows a completely different aspect of stream channels in this reach: channel morphology before 1976 is different from that after 1976. This difference is represented by the two different trends of the data. The authors should run non-linear regression to establish power functions for the two different trends in each listed month and then link this difference to the possible difference of vegetation cover in the two different time periods. This would strengthen the analysis a lot. Fig. 10 is not well tied to the data shown in Figs.8 and 9. It is nice, but there lacks evidence to support it.

**Response: OK, it is very good suggestion. We adopt non-linear regression to build power functions in each listed month and link these differences to the possible difference of vegetation cover. Moreover, we rethought Fig.10 and augment the analysis on Fig.9.**

Section 5.1.2: First, the authors should mentioned Fig. 11 first and then Fig. 12.

**Response: OK. We have corrected this.**

Second, the big problem here is that the postulation raised here (lines 15-19 on page 11) is not fully supported by the only data shown in Fig. 11. The stage data in Fig. 11 are not sufficient to argue the change of flow regime exactly because the channels here are anabranching channels. This means that the same flow stage in different seasons might be associated with different water discharges. Maybe there are no water discharge data available in this reach. If this is the case, the authors should re-think their arguments: the fact that these channels are stable means that sediment (bedload) supplied from upstream (i.e., the Dari reach) is balanced by the sediment transport capacity in this reach. One way might be useful is to compare the supplied bed load based on the prediction made for the Dari reach with the transport capacity predicted in this reach. The authors should expect that they are similar or very close to each other. Then, the impact of vegetation on the hydraulics might be reflected in Manning's n used in the bedload model. Comparing this value with the one used in the Dari reach may show the impact of vegetation on the stable status of this reach.

**Response: We agree with the detailed analysis above. Since Fig.11 did not fully support our analysis, we continue to collect the data of monthly-channel discharge and monthly-sediment transport rate in four hydrological stations (Huangheyan, Dari, Maqu, Tangnaihai). New data and analyses can strengthen this section, in particular, the impact of vegetation more distinct.**

Section 5.1.3: This reach is a tributary. If the authors have no water discharge data in this tributary, I suggest to delete this part completely from the current manuscript. This is because only showing a postulated diagram (i.e. Fig. 13) is insufficient to convince the readers about the status of this reach.

**Response: OK. In section 5.1.3, the Lanmucuo River is a small meandering river which has no hydrological data, but we conducted field investigations during 2011-2015. Especially, in 2015 we measured the cross-section and mean velocity in the middle reach. Perhaps we delete the Fig. 13, but add other data or figure so as to keep the integrity of this study.**

Section 5.1.4: This reach is unstable. Again, just using the temporal changes of channel sections between three years (i.e., Fig. 15) is not enough to explain how vegetation affects them. Again, I think it is very important for the authors to predict bedload transport rates and then use them to calculate the mean sediment load in this reach. By comparing this (or these) mean value(s), the authors may argue that why the reach is not stable. In the meantime, comparing the value of manning's n used in this reach with those used in the first and second reaches along the main river would provide evidence of the impact of vegetation on river morphology.

**Response: Yes, the braided reach of Daheba River is quite unstable and the vegetation effect can be ignored here. Actually, the authors have predicted bedload transport rates in this reach. Unfortunately, there are no measured data of bedload transport rates for comparison.**

Minor points:

Lines 5-8 on page 6: this description is very confusing;

**Response: OK. We have revised it.**

Lines 17-18 on page 7: why should the stable reach have high bedload transport capacity?

**Response: The reach is very stable because the dense trees develop on bars/islands as well as river banks. If over-capacity bed load is incoming, the reach is very stable because trees densely develop on bars/islands. If over-capacity bed load is incoming, the**

**stable anabranching channel can not be widened and keep high velocity within the channel so as to efficiently transport bedload relative to unstable braided channel.**

Lines 4-5 on page 9: what does the rivers in an arid area have anything to do with rivers in the study area?

**Response: Here we cited the references for arid area to justify the correctness of using the Manning formula as flow resistance.**

Figure 1: Please mark R1, R2, R3, and R4. Also, only use the arrow to show the direction of flow.

**Response: OK, we add R1,R2, R3, R4 in Fig.1 and correctly use the arrow.**

In the legend, 'Tributary' and 'Trunk stream' should be reversed. Please use 'Main stream' rather than 'Trunk stream';

**Response: Yes, I have changed it immediately.**

Figures 3-6: these figures should be combined into one figure;

**Response: No problems. We are able to combine them into one figure.**

Figure8: please use the same legend for the two figures;

**Response: OK, I have revised it quickly.**

Figure 14: it does not help much in understanding the difference between the regular and flood conditions;

**Response: We have chosed this figure and better images.**

**Response to Rereree #2**

**We appreciate Prof. Coenders-Gerrits M. (Referee #2)'s comments and suggestions. These comments are very used to enhance the manuscript during the following revision. A point-by-point responses to each comment are addressed below.**

The authors present a relevant study on the effect of vegetation on bedload transport capacity and channel stability. Therefore, they study 4 reaches of the upper Yellow River, China. The 4 reaches differ in planform. Despite the potential interest, the paper is highly descriptive and hypothetical. Barely any data is collected to justify the conclusions. This leads to the question what we can learn from this study. The river planform is not really something we can easily adjust and the role of vegetation is more a result of the planform, than a cause. Maybe this also relates to the fact that there is no study objective given.

**Response: Many thanks for your objective remarks about the manuscript. We confess that the interesting phenomenon in the Yellow River source needs more data to verify our conclusion. This study about river planform of the Yellow River source is an intriguing but understudied part of the world – altitude, plateau landscapes, and its global significance, so we need strong foundation studies to set up further analyses-given data limitations, these will be inherently descriptive in the first instance, but it is important to get this right. We still believe the role of vegetation plays a great role on the planform in this region, though there is a lack of direct evidence. Perhaps we need to go further in making relations to other parts of the world, in terms of the influence of landscape and environmental setting upon river diversity that these relations are the same here, or there are some notable differences.**

Abstract:

The abstract starts immediately with describing what the study entails, but the existing knowledge gap is missing. As well as the 'reason for this study'.

**Response: It is a very good suggestion. We add 1-2 sentence to explain the existing knowledge gap missed and the reason of this study.**

Introduction:

The introduction is really long and very general. It seems like a 'lecture' on river planforms in relation to bars. I would advise to shorten the introduction and focus on what is currently missing (knowledge gap) and why this study is relevant (what will it bring). Furthermore, I would also explain how the existing study differ from exiting studies.

**Response: OK, we are pleased to accept this valuable advice to compress the introduction. The knowledge gap has been seriously considered and answer why this study is relevant and differs from existing studies.**

P9L1-25:

Add dimensions or units to symbols

**Response: OK, I can do it.**

Equation 3-5:

Why do you need Eq. 4 if you can also derive it from Eq. 3 and 5?

**Response:Definitely Eq.4 is derived from Eq.3 and Eq.5. Eq. 3 gives us the dimensionless bedload transport rate per channel width, but we want to obtain the dimensional bedload transport rate per unit channel width. So keeping Eq.4 in text is reasonable.**

Section 5:

Based on what can the authors conclude how the bars are developed/eroded? (fig 10, 12,13). Can this not better be answered with satellite images over several years?

**Response:Figure 10, 12, 13 are simple sketches of the bars in braided, anabranching and meandering channel based on our field investigation and satellite images. Adopting the satellite images is a good option, but the difference of water depth in the different satellite images so that the submerging range in channel varies. After discussing with other authors, we w seriously considered the availability of satellite images in this study.**

Figure 1:

Naming R1, R2, R3, and R4 are not visible in the figure

**Response: OK, I can do it.**

Figure 8:

What's happening during the low flows? This seems to weird behaviour. How can the stage drop when Q increases? That is remains constant is possible if the river width increase after a certain threshold, but this seems unrealistic. Please elaborate/explain.

**Response:Your questions make sense. We believe the data is correct. During the low flows, the channel partly is frozen in December, January, February, March, and April. Because the water**

**in lower layer is frozen, the stage of incoming flow increases but the discharge still very lower or keep constant. Therefore, in the low flows, the stage increases when Q is nearly constant.**

Figure 9:

Please be consistent. The upper graphs are Qh-plots, while the lower two are hQ-plots.Furthermore, the coloring is not that clear, which makes the plot difficult to interpret.

**Response: Many thanks for pointing out this mistake. The upper and lower graphs are Q-h plots, but the coordinate texts of the lower graphs are wrong. Meanwhile, we adopt Adobe Photoshop CS to processing the coloring image by increasing the resolution.**

Figure 11:

Is the stage unit correct? What is the datum of this stage?

**Response:The stage value is correct. The datum of this stage is the elevation of water surface. We will double-check the data and add the explanation in data source avoiding the misunderstanding.**

Throughout the entire manuscript:

• Textual: after "i.e." and "e.g." a comma should be placed

**Response: Good point! I have added a comma for all** "i.e." and "e.g."**.**

• Order appearance figures in text, is order figure numbers (e.g., figure 11 and 12).
Please check

**Response:OK. I have updated the figures order in text.**

**Reply to the editor**

We deeply appreciate the constructive comments of the reviewers (Prof. Peng Gao and Prof. Coenders-Gerrits M.) and the major revisions from the Editor (Prof. Günter Blöschl) on our manuscript of '**hess-2015-526**'. These suggestions are quite helpful for us and we have incorporated all comments into the revised manuscript. During the last two month, we have revised the paper so as to greatly improve the quality. We have replied all comments point-by-point. The revised words, sentences and references of the manuscript were highlighted in red color in the marked manuscript. Meanwhile, the reason and explanation will be addressed below one by one to the major revisions. The authors would like to continue to polish the paper until reaching the level of publication.

The major revisions are addressed below.

1. The authors add two sentences to address the existing knowledge gap in the Abstract and Introduction. Meanwhile, we shorten the Introduction about the bar, following the suggestion of the reviewers.

2. We have run non-linear regression (power function) for the data of the water stage and discharge in June, July, August, and September in 1966 to 1984, and run the regression analysis in July from 1964-1984, and accordingly analyze the difference of the coefficients and exponents in different periods. R2 of all regression analysis is up to 0.98.

3. We slightly adjusted the parameters of Maqu and Daheba reaches in Table 3 after discussing with co-authors.

4. Revised Figure 1 and combined the previous Figures 1 and 2 into Figure 1.

5. Figures 3-6 are combined into Figure 3.

6. Revised the legend of Figure 8.

7. Add a comma after "i.e." and "e.g."

8. Double-check the list and citation of references.

The minor revisions are addressed below.

1. Page 1, add a new affiliate in Line 6-7, and add a sentence in the Abstract in Line 23-25.

2. Page 2, revised the sentence in Line 2-3, and add a sentence in 22-23.

3. Page 5, add a citation in Line 14, and "previous" in Line 18.

4. Page 6, revise the sentence in Line 3-6, and add two citation in Line 9 and 13.

5. Page 7, revise the number of figures in Line 6-7, 14-15, and 31. And add "1.5-3.0 km wide" in Line 13.

6. Page 8, revise the number of figures. Add "median size" in Line 13, " the" in Line 23.

7. Page 9, add the unit for all parameters. Add "Daheba reach", "Lanmucuo River", "Dari River", and " Maqu reach" in Line 9-11.

8. Page 10, revise the number of figures in Line 9 and 19. Add the regression analysis in Line 25-31.

9. Page 11, revise the number of figures in Line 4-5, 23-24. Add the regression analysis in Line 9-14 and 26-32. "occurred" was replaced by "changed" in Line 17.

10. Page 12-13, revise the number of figures. And revise two sentences in Line 14-18.

11. Page 15, revise the acknowledgements.

[revised manuscript text omitted]